# COVID-19-Related Vaccine Hesitancy among Community Hospitals’ Healthcare Workers in Singapore

**DOI:** 10.3390/vaccines10040537

**Published:** 2022-03-30

**Authors:** Junjie Aw, Sharna Si Ying Seah, Benjamin Jun Jie Seng, Lian Leng Low

**Affiliations:** 1Outram Community Hospital, SingHealth Community Hospitals, 10 Hospital Boulevard, Singapore 168582, Singapore; sharna.seah.s.y@singhealthch.com.sg (S.S.Y.S.); low.lian.leng@singhealth.com.sg (L.L.L.); 2SingHealth Duke-NUS Family Medicine Academic Clinical Program, Outram Rd, Singapore 169608, Singapore; 3MOH Holdings Singapore, 1 Maritime Square, Singapore 099253, Singapore; benjamin.seng@mohh.com.sg; 4Department of Family Medicine and Continuing Care, Singapore General Hospital, Outram Rd, Singapore 169608, Singapore; 5SingHealth Regional Health System PULSES Centre, Singapore Health Services, Outram Rd, Singapore 169608, Singapore

**Keywords:** COVID-19, coronavirus disease-19, COVID-19 pandemic, SARS-CoV-2 infection, 2019 novel coronavirus disease, vaccines, COVID-19 vaccines, vaccine hesitancy, barriers, vaccine acceptance, associations, healthcare workers, knowledge, attitudes, perceptions, qualitative, Singapore

## Abstract

COVID-19 has culminated in widespread infections and increased deaths over the last 3 years. In addition, it has also resulted in collateral economic and geopolitical tensions. Vaccination remains one of the cornerstones in the fight against COVID-19. Vaccine hesitancy must be critically evaluated in individual countries to promote vaccine uptake. We describe a survey conducted in three Singapore community hospitals looking at healthcare workers’ vaccine hesitancy and the barriers for its uptake. The online anonymous survey was conducted from March to July 2021 on all staff across three community hospital sites in SingHealth Singapore. The questionnaire was developed following a scoping review and was pilot tested and finalized into a 58-item instrument capturing data on demographics, contextual features, knowledge, attitudes, perceptions, and other vaccine-related factors in the vaccine hesitancy matrix. Logistic regression analysis was employed for all co-variates that are significant in univariate analysis. The response rate was 23.9%, and the vaccine hesitancy prevalence was 48.5% in the initial phase of the pandemic. On logistic regression analysis, only being female, a younger age, not having had a loved one or friend infected with COVID-19 and obtaining information from newspapers were associated with vaccine hesitancy in healthcare workers in Singapore community hospitals.

## 1. Introduction

Globally, the ongoing 2019 (COVID-19) coronavirus pandemic has culminated in 319 million infections and 5.52 million deaths as of 14 Jan 2022 [1]. It has also impacted global economies negatively and created new challenges in healthcare as well as geopolitical relations between countries [2]. Notably, global economies have been projected to slow down by 2%, while global trade is expected to decrease by 13–32% [3]. In addition, unprecedented restrictions in international and domestic travel have also been implemented during the course of the pandemic with lockdowns, especially in countries coping with high COVID-19 caseloads.

The impacts of COVID-19 within Singapore mirror those globally. Initially we drew on the lessons learned from the SARS outbreak in 2003. Non-contact temperature- screening checkpoints were swiftly set up, with border control instituted immediately. But we soon realized that temperature screening wasn’t effective [4]. A multi-ministry task force was set up in consultation with the Director Medical Service (Ministry of Health) to coordinate public policies and strategize planning. An aggressive policy of swabbing, isolating and contact tracing to ring-fence infection clusters was pursued until a few months ago in a national effort to contain the numbers for hospitals to continue to function smoothly. New concepts of COVID-19 treatment facilities (CTFs) and COVID-19 Care Facilities (CCFs) were birthed in which community facilities were repurposed from existing infrastructure nationwide to provide care based on risk stratification of COVID-19 patients.

Circuit breaker, Singapore’s terminology for lock-down, was flipped on and off in parallel with monitoring of epidemiology of the pandemic waves in Singapore. Institution and enforcement of other social measures such as mask wearing and social distancing in malls and restaurants, coupled with a successful vaccination drive, prevented Singapore’s healthcare system from being overwhelmed [5]. Nonetheless, even in June 2020, COVID-19 had already increased burnout of nurses and doctors in Singapore [6].

The COVID-19 disease also constitutes a historical pandemic where widespread dissemination of information is conducted through the use of technology and social media [7]. The overwhelming volume of scientific information on COVID-19 disease and its complications has facilitated and amplified an infodemic. An infodemic is defined by the World Health Organization as “excessive information including false or misleading information in digital and physical environments during a disease outbreak.” [8]. Importantly, a recent systemic review has shown that misinformation related to COVID-19 is present in up to 28% of social media posts [9]. Misinformation related to COVID-19 has been shown to have a negative impact on public health efforts to slow the spread of disease and increase uptake of vaccination.

Health literacy related to COVID-19 disease has been shown to be highly variable, even among healthcare workers [10]. Among the local general population, a study conducted at the initial phase of the pandemic showed relatively high levels of COVID-19 knowledge [11]. There remains a paucity of data pertaining to COVID-19-related perceptions and attitudes and their association with vaccine hesitancy rates among healthcare workers in Singapore.

In this paper, we aim to report the prevalence of vaccine hesitancy and its associated barriers among healthcare workers in hospitals in Singapore.

## 2. Methodology

### 2.1. Study Design and Institutional Review Board Approval

An online survey was conducted in Singapore from 19 March 2021 to 2 July 2021 among healthcare workers working in SingHealth Community Hospitals (SCH), an institution comprising of Outram Community Hospital, Sengkang Community Hospital and Bright Vision Community Hospital. The three hospitals are located within the largest healthcare cluster in Singapore [Singapore Health Services (SingHealth)] and they provide a full suite of medical, nursing and rehabilitation services for patients who require step-down care from acute hospitals [12]. The survey methodology and results were reported in accordance with the Checklist for Reporting Results of Internet E-Surveys (CHERRIES) [13]. 

A 58-item instrument was used in this study (Appendix A). Of the 58 items in the survey, only 7 items were optional, and of these, 2 were free text responses that elicited reasons for not undertaking COVID-19 vaccination in the initial drive and concerns on long-term side effects from COVID-19 vaccination. The survey used a secure anonymous online survey platform developed by Government Technology Agency of Singapore (FormSg) [14]. Consent was implied when participants clicked on the email invitation link or scanned the QR code. Study participants were informed of the length of the survey and the data storage procedure prior to starting the survey. No personal data was collected in this anonymized survey.

The study protocol was exempted from review and approved by SingHealth Institutional Review Board iSHaRe (Application number 2021/2753).

### 2.2. Development and Pre-Testing

SCH utilized an instrument with four key domains related to COVID-19 that included knowledge, perception, attitudes and a COVID-19 vaccine hesitancy matrix. Items used in the domains of knowledge, attitude and perception related to the COVID-19 pandemic were identified from various questionnaires in a systematic review as well as the Brief Illness Perception Questionnaire [10,15]. Items used in the COVID-19-related vaccine hesitancy matrix were identified from a systematic review [16]. Vaccine hesitancy is defined by the World Health Organization as the delay in acceptance or refusal of vaccines despite the availability of vaccination programs [17]. Our participants were considered to be vaccine hesitant if they did not agree to being vaccinated in the initial call for COVID-19 vaccination.

An initial 66-item questionnaire was developed for the study. It was pilot-tested among a group comprised of ten independent healthcare administrators, nurses, allied health staff and medical doctors. In the pilot exercise, the questionnaire was self-administered, and retrospective probing was used to identify any item which was unclear to the study participants. After deliberation with the study investigators and participants involved in the pilot, 8 items with duplicated themes were removed to reduce participants’ fatigue.

The instrument was developed and administered in English only, as our participants were all literate in English. The average time taken to complete this questionnaire was 15 min.

### 2.3. Recruitment Process and Access to Questionnaire

The link and QR code of the online survey were sent out to healthcare workers via the hospitals’ internal email network. Reminder emails to participate in the study were sent every week for a total of 14 times. The survey was also shared in departmental meetings and advertisements were placed in the form of posters around various departments within the three hospitals to increase awareness. In addition, invitation links were also posted on “Workplace by Meta”, an online social media platform connecting staff to increase awareness and encourage uptake.

### 2.4. Sample Size Computation

OpenEpi version 3.01 was used to calculate the sample size. The minimum sample size was 225 for a population of 1008 healthcare workers across SingHealth Community Hospitals with an estimated hesitancy of 25% (vaccine acceptance at 75%), a confidence level of 95% and a margin of error of 5%.

### 2.5. Survey Administration

FormSG was used to capture the survey results. Each participant’s responses were automatically captured into the FormSG database upon completion. A dedicated link and a QR code to the online survey were created for the purposes of this study and participation was voluntary. Implied consent was deemed to have been given if participants clicked on the link or scanned the QR code. There was no monetary or non-monetary incentive given for the completion of the survey.

Randomization of questions was not performed, as items in questionnaires were grouped by respective domains for ease of readability. The items were displayed in a continuum in a single webpage and built-in prompts for incomplete items were used for completeness checks. Due to limitations within the survey platform, participants were not able to review or change their answers after submission of their response.

### 2.6. Response Rates and Preventing Multiple Entries from the Same Individual

The response rate was computed by dividing the number of study participants over the number of healthcare workers within the three community hospitals. To prevent multiple entries from being submitted from the same individual, the use of cookies was used to assign a unique identifier to each entry.

### 2.7. Data Analyses

In the study, only completed questionnaires were analyzed. Study participants who terminated the survey without completing the form were deemed to have withdrawn their consent. Such data was not captured nor analyzed. Due to platform limitations, it was not possible to capture the time each participant took to complete the form.

Descriptive statistics were used to characterize the study population and study responses. Categorical variables were presented as proportions and continuous variables summarized as medians (25th and 75th percentiles) or means (standard deviations), as appropriate. A Pearson chi-square test or Fisher’s exact test (for cell counts less than or equals to 5) was used to compare categorical variables. A Mann–Whitney U test was used to compare ranks for non-normally distributed continuous variables, whereas the 2-tailed Student t-test was used to compare ranks for normally distributed, continuous variables where appropriate.

Logistic regression analysis was further performed for all co-variates that were significant on univariate analysis. 

A two-tailed *p* value < 0.05 was considered statistically significant and statistical analyses were performed using Stata version 13.0.

A narrative description of themes identified from the 2 optional free text responses were performed by J Aw and SSY Seah for congruence. Any disagreements were internally mediated between the 2 authors and a final agreement decided upon after the mediation.

## 3. Results

Among the 1008 employees across SCH, a total of 241 responses were captured and the prevalence of COVID-19 vaccine hesitancy was 48.5%.

Table 1 describes the demographics of the participants. Males (*p* = 0.003), older participants (*p* = 0.022) and non-citizens (*p* = 0.022) were less vaccine hesitant. When grouped into those age 40 and younger against those age 41 and above, the former was more vaccine hesitant, with the difference almost approaching statistical significance (*p* = 0.054).

Among the professions in the hospitals, physicians and nurses were less vaccine hesitant (*p* < 0.001).

Participants staying with colleagues, friends or housemates compared to other living arrangements (*p* = 0.007) were less likely to be vaccine hesitant.

Ethnicity had a statistical association with vaccine hesitancy when grouped by Chinese and non-Chinese (*p* = 0.048), but the association was lost when ethnicity was described in its usual subgroups in the Singapore context.

Income levels, educational levels, marital status and the number of children in a family unit were not associated with vaccine hesitancy.

Table 2 describes the associations with vaccine hesitancy as per vaccine hesitancy determinants matrix recommended by SAGE Working Group on Vaccine Hesitancy [17].

### 3.1. Contextual Determinants of Vaccine Hesitancy

Participants who rated hospital leaders highly on the effectiveness of their communication to encourage vaccine uptake were more accepting of the vaccine (*p* = 0.046; Table 2 and Appendix A). 

Participants who had loved ones or friends infected with COVID-19 were less likely to be vaccine hesitant (*p* < 0.001).

On communication channels, participants who obtained COVID-19 vaccine information from family and friends (0.023) were more vaccine hesitant and those who did so from newspapers sources (*p* = 0.024) were surprisingly so, too.

By contrast, those who obtained COVID-19 vaccination via personal opinions and forums on social media were not associated with vaccine hesitancy. Exploratory analysis led to the observation that participants who obtained COVID-19 vaccine information from such sources also engaged in a higher number of channels for information gathering (median channels = 5 [interquartile range 3,7]) compared to those who did not (median channels = 3 [interquartile range 2,5]) and the difference is highly significant statistically (Appendix A; *p* < 0.001).

Participants who obtained their COVID-19 vaccination information via credible authoritative websites or via credible social media homepages (e.g., Ministry of Health’s Facebook or Instagram) had no association with vaccine hesitancy.

The perceived clarity in organization communication on all matters related to COVID-19 and its vaccination was equally high in both vaccine hesitant and vaccine accepting groups and was not statistically associated with vaccine hesitancy.

### 3.2. Individual and Group Influences

Participants who believed in herd immunity even without vaccinating were more vaccine hesitant (*p* = 0.003), whereas those who were more confident of the vaccine’s ability to protect themselves (*p* = 0.003) or their loved ones and friends (*p* = 0.004) from COVID-19 after vaccination were less vaccine hesitant. 

Interestingly, participants who felt coerced into taking up the vaccination were three times more likely to be vaccine hesitant (*p* = 0.006).

On trust, participants with a lower perceived transparency on information conveyed by the authorities about safety of COVID-19 vaccines were more vaccine hesitant (*p* = 0.002). Trust in Singapore’s health ministry in decision making for country’s best interest and in the healthcare system in handling an outbreak ran high in both vaccine hesitant and accepting groups and had no statistical significance with vaccine hesitancy.

Participants who rated higher in self-knowledge related to COVID-19 were more accepting of the vaccination, although the difference did not reach statistical significance (*p* = 0.054). 

Among the risk–benefit matrix, participants whose work involved direct contact with COVID-19 patients was the only statistically significant association with vaccine acceptance (*p* < 0.001).

The presence or absence of voluntary uptake of vaccination in the past was not associated with vaccine hesitancy, as most of the healthcare workers in SCH had voluntarily undertaken vaccinations in the past (73.4%).

### 3.3. Vaccination-Specific Determinants of Vaccine Hesitancy

Participants who expressed an increased likelihood of taking a vaccine when offered free and within easy reach were associated with vaccine acceptance (*p* < 0.001).

With regards to long-term concerns on COVID-19 vaccination, there were 227 responses out of 241 participants, representing a response rate of 94.2%. Of the 14 who did not respond, 1 had no-sense input and 13 had inputs “.” or “-”.

Those who had long-term side effects concerns were more vaccine hesitant (*p* = 0.016). Sensitivity analyses for the missing data showed the association remained significant, indicating robustness of the results (Table 2).

Other concerns on vaccine characteristics, such as its efficacy, etc., had no significant association with vaccine hesitancy statistically.

Participants who did not believe in vaccination were the minority in SCH (n = 6, 2.5%). Of these, 83.3% chose not to accept COVID-19 vaccination, although the difference between the groups did not reach statistical significance (*p* = 0.111).

Table 3 summarizes the unadjusted odds ratio for all the significant associations with vaccine acceptance on univariate analysis and adjusted odds ratio after logistic regression analysis.

Age (1.06 [1.02–1.10]), males (2.44 [1.03–5.78]), participants with knowledge of loved ones or friends infected with COVID-19 (4.9 [1.68–14.27]) and those who obtained COVID-19 vaccine information from newspapers (0.34 [0.16–0.72]) retained statistically significant associations with vaccine acceptance.

### 3.4. Qualitative Results from Free Text Inputs

Qualitative free text responses to the following two questions (reflected in Appendix A):

“If you did not sign up for the COVID-19 vaccination during the first call for vaccination, can you tell us why?” and

“Do you have any concerns with regards to the long-term consequence(s) of the COVID-19 vaccination? Can you share with us if you have any concern(s)?”

In the 1st qualitative free text responses (Appendix A), a total of 79 responses were recorded among 117 who rejected the call for vaccination in the 1st round. Of these, 2 had no-sense text inputs and a resultant 77 participants’ responses were recorded with a net total of 101 datapoints observed. Agreement between the 2 authors J Aw and SSY Seah was 87.1% for the 101 datapoints. A total of 11 themes with 1 miscellaneous group were crystallized and the top 3-most-frequent reasons for declining the 1st round of call up for COVID-19 vaccination were due to side effects in general (n = 24 [23.7%]), lack of information about COVID-19 vaccination (n = 15 [14.8%]) and concerns over allergies due to vaccination (n = 11 [10.9%]).

In the 2nd qualitative free text responses, a total of 92 participants out of 241 gave feedback. A net total of 18 themes and 1 miscellaneous group were crystallized (Appendix A). The agreement between J Aw and SSY Seah was 93.7% for a total of 127 datapoints captured. The top three themes cited were due to general long-term side effects (n = 25 [19.7%]), family planning, pregnancy, breastfeeding-related concerns, effects on offspring (n = 17 [13.4%]) and fear of the uncertainty in future side effects (n = 16 [12.6%]). 

## 4. Discussion

To the authors’ knowledge, this is the first paper evaluating vaccine hesitancy and its associations among healthcare workers in Singapore. Overall, the prevalence of vaccine hesitancy was comparable to other developed countries in the initial phase of the COVID-19 pandemic [18,19,20,21,22,23,24,25,26].

Our paper adds to prevailing global evidence on how socio-demographic factors, younger age and female sex, specifically, are associated with COVID-19 vaccine hesitancy among healthcare workers, which mirrored findings from studies conducted in the general population [27,28,29,30]. It is interesting to note that similar patterns existed in earlier studies of vaccine hesitancy to other vaccine types even before the COVID-19 pandemic, with those who are younger and females being more hesitant towards vaccination [31,32].

Gender-related differences in COVID-19 vaccine hesitancy might be due to reasons related to family planning, breastfeeding and concerns over long-term effects the vaccine may have on participants’ offspring, as illustrated in the analysis of the qualitative inputs (Appendix A). Of note, the COVID-19 infection was associated with significant adverse outcomes in pregnancy with increased risk of preeclampsia, preterm and stillbirth, wheras COVID-19 vaccination was not associated with increased vaccine-related adverse events or poorer obstetric/neonatal outcomes [33,34]. It is important to release safety and efficacy data of vaccine in future pandemics rapidly to allow pregnant and breastfeeding women to make informed decision for vaccination. Alignment of statements among global medical authorities and organizations with formation of an unified consensus build confidence and trust in reassuring and emphasizing the safety and efficacy of COVID-19 vaccination during pregnancy and breastfeeding [35]. 

On age, one qualitative study found that young adults lacked complete information, as most were dependent on news streamed to their phones and their main concerns were on long-term side effects of the vaccine [36]. Another plausible reason for the hesitancy associated with younger age might be due to their relatively healthier sense of well-being with lack of all the co-morbid medical conditions which had been shown to increase morbidity and mortality with COVID-19 infection [37,38]. Yet another reason could be the inverted-U pattern of risk-taking behaviours across the age spectrum, peaking in late adolescence to early adulthood [39]. Hence, reaching out to this demographic group with vaccine hesitancy will require targeted messaging on COVID-19 risks for individuals without co-morbidities while at the same time conducting research on what appeals to this group, be it the social cohesiveness concept of protecting their loved ones or the socio-economic benefits of being able to socialize and secure an economic future [40].

There is currently a dearth of qualitative research exploring deeply these differences towards vaccination by sex and age across different geopolitical regions and culture. It may be important to address this research gap in future qualitative research.

In this study, having past experiences with a loved one or friend being infected with COVID-19 was shown to be associated with reduced vaccine hesitancy. The recruitment of COVID-19 patients and their family members as vaccine ambassadors for vaccine uptake could potentially provide novel narratives to aid in vaccination drives and target misinformation in the pandemic [41,42].

Surprisingly, the use of a traditional media source such as the print newspaper for COVID-19 vaccination information was associated with vaccine hesitancy in our context. This differed from studies in the United States, where the use of traditional information such as newspaper was associated with increased vaccine acceptance [43]. The reason for this is worth further exploration in future studies.

Interestingly, we also did not find a significant association between vaccine hesitancy and consumption of social media comprised mainly of personal opinions, unlike other sources [44,45]. We think that this could be due to the minority status of our participants who obtain information regarding COVID-19 vaccination from such a channel. The second possibility might be because these same participants were obtaining COVID-19 vaccine information from numerous other channels compared to those who did not declare obtaining information from such private social media channels. This observation may be important to explore in future studies looking at designing communication models to prepare for the next fight in misinformation in future pandemics [30].

The research gap in the complexities and role of mass media needs further exploring, be it in the form of targeted messaging, or the manner and channel the messages are broadcast, to overcome some of the vaccination barriers [46,47,48,49,50].

In our context, there was a high level of perceived trust among healthcare workers in Singapore’s ministry of health and in the ability of our healthcare system to cope [51]. Such high levels of trust could have interacted with the other known factors with vaccine hesitancy and lowered their impact on vaccine hesitancy.

### Limitations

The study findings should be interpreted in the context of these limitations. Firstly, we were not able to establish a direct causal link between studied variables and vaccine hesitancy, as this was a cross-sectional paper.

Secondly, while this survey was adequately powered for evaluation of vaccine hesitancy prevalence, assessment of vaccine hesitancy associations with some co-variates may be underpowered, leading to Type 2 errors and reduced generalizability of the results. Larger studies are required to confirm our study findings.

Lastly, as our survey was conducted in the earlier part of the pandemic, there could be new determinants of vaccine hesitancy in a prolonged pandemic with pandemic fatigue and need for booster vaccination.

Nonetheless, our findings serve as an important foundation for future studies to address these research gaps, now or in the next pandemic, where it is not unlikely a new vaccine will be designed for mass vaccination.

## 5. Conclusions

The prevalence of vaccine hesitancy was high within healthcare workers in Singapore during the initial stage of the COVID-19 pandemic. Although underpowered, we found that the risk factors associated with vaccine hesitancy included younger age, being female, obtaining COVID-19-related information from newspapers and having loved ones or friends who had not yet contracted COVID-19. Age and sex associations with COVID-19 vaccine hesitancy are comparable to global data and to past studies for vaccine hesitancy for other vaccine types. They may be important factors for policymakers to consider in formulating campaigns for vaccine uptake. Vaccine hesitancy is complex and involves an interplay of factors. Mitigation measures to increase uptake depend on adopting a multi-pronged and multi-system approach to overcome barriers systematically based on findings unique in each country associated with hesitancy.

## Figures and Tables

**Table 1 vaccines-10-00537-t001:** Sociodemographic profiles of participants.

Sociodemographics	Total (n = 241)	Vaccine Hesitant (n = 117)	Non-Vaccine Hesitant (n = 124)	*p*-Value
Male, n (%)	50 (21)	15 (13)	35 (28)	0.003
Age ^~^, median (25th, 75th percentiles)	33 (29, 40)	32 (28, 38)	34 (30, 43)	0.022
Age groups, n (%)				0.054
*21–40 years*	184 (76)	96 (82)	88 (71)	
*41 and above years*	56 (23)	21 (18)	35 (28)	
Profession, n (%)				<0.001 *
*Administrative*	63 (26)	42 (36)	21 (17)	
*Allied Health*	81 (34)	44 (38)	37 (30)	
*Physician*	37 (15)	12 (10)	25 (20)	
*Nurses*	60 (25)	19 (16)	41 (33)	
Income, n (%)				0.053 *
*SGD 2999 and below*	99 (41)	43 (37)	56 (45)	
*SGD 3000-4999*	74 (31)	43 (37)	31(25)	
*SGD 5000-7999*	42 (17)	23 (19)	19 (15)	
*SGD 8000 and above*	26 (11)	8 (7)	18 (15)	
Nationality, n (%)				0.022
*Non-Citizens*	79 (33)	30 (26)	49 (40)	
*Citizens (Singaporeans)*	162 (67)	87 (74)	75 (60)	
Ethnicity, n (%)				0.048
*Chinese*	158 (66)	84 (72)	74 (60)	
*Non-Chinese*	83 (34)	33 (28)	50 (40)	
Ethnicity, n (%)				0.102 *^#^
*Chinese*	158 (66)	84 (72)	74 (60)	
*Indian*	9 (4)	4 (3)	5 (4)	
*Malay*	12 (5)	7 (6.0)	5 (4.0)	
*Others*	62 (26)	22 (18.8)	40 (32.3)	
Marital status, n (%)				0.315 ^#^*
*Single*	106 (44)	49 (42)	57 (46)	
*Married*	130 (54)	67 (57)	63 (51)	
*Divorced*	2 (1)	1 (1)	1 (1)	
*Widowed*	3 (1)	0 (0)	3 (2)	
Number of child(ren), median (25th, 75th percentiles)	0 (0, 2)	0 (0, 2)	0 (0, 2)	0.834
Has at least 1 child	106 (44)	51 (44)	55 (44)	0.905
Education level, n (%)				0.075 ^#$^
*Primary school, GCE ‘N’ and ‘O’ levels, and below*	14 (6)	3 (3)	11 (9)	
*Diploma and GCE ‘A’ levels*	48 (20)	27 (23)	21 (17)	
*Bachelor’s degree and above*	179 (74)	87 (74)	92 (74)	
Living arrangement, n (%)				0.007 ^$^
*Staying alone*	15 (6)	7 (6)	8 (6)	
*Staying with family*	182 (76)	98 (84)	84 (68)	
*Staying with friends, colleagues and/or housemates*	44 (18)	12 (10)	32 (26)	

^~^ missing data n = 1; * considered significant if *p* < 0.008; ^#^ Fisher exact test applied; ^$^ considered significant if *p* < 0.0167.

**Table 2 vaccines-10-00537-t002:** Vaccine hesitancy matrix.

	Vaccine Hesitant (n = 117)	Non-Vaccine Hesitant (n = 124)	*p*-Value
**Contextual features**			
Presence of loved ones or friends infected with COVID-19, n (%)	10 (9)	32 (26)	<0.001
Perceived effectiveness of SCH leadership influence on vaccine uptake, median (25th, 75th percentiles) (0 = Not effective at all, 10 = Extremely effective)	8 (7, 9)	8 (7, 9)	0.046
**Communication**			
Perceived clarity of organization’s communication, median (25th, 75th percentiles) (0 = Not clear at all, 10 = Extremely clear)	8 (7, 9)	8 (7.5, 10)	0.30
Sources of COVID-19 vaccine information, n (%)			
*SCH’s virtual townhalls*	83 (71)	78 (63)	0.19
*Managers and supervisors*	65 (56)	65 (52)	0.63
*Family and friends*	26 (22)	14 (11)	0.023
*Scientific journals*	46 (39)	42 (34)	0.38
*Credible websites (e.g., World Health Organization and Ministry of Health’s websites)*	62 (53)	58 (47)	0.34
*Print Newspaper*	44 (38)	30 (24)	0.024
*Credible social media pages (e.g., Ministry of Health’s Facebook or Instagram)*	58 (50)	49 (40)	0.12
*Personal opinions and forums on Social Media e.g., posts from influencers*	20 (17)	14 (11)	0.20
*Self-declared lack of reading on COVID-19 vaccination*	3 (3)	1 (1)	0.36
**Individual and Group influences**			
**Beliefs, attitudes about health and prevention, median (25th, 75th percentiles)**			
Perceived helpfulness of protective measures in protecting self from getting infected. (Protective measures include social distancing, wearing face mask, practicing proper hand hygiene and staying at home as much as possible and excludes receiving COVID-19 vaccination.) (0 = Not helpful at all, 10 = Extremely helpful)	8 (7, 9)	8 (7, 9)	0.54
Agreement with the COVID-19 regulations (e.g., safe-distancing and mask wearing) in place. (0 = Fully disagree, 10 = Fully agree)	10 (8, 10)	10 (8, 10)	0.91
Belief in achieving herd immunity towards the COVID-19 virus even if he/she does not get vaccinated. (0 = Fully disagree, 10 = Fully agree)	5 (2, 7)	3 (1,6)	0.003
Perceived possibility of contracting COVID-19 virus without being vaccinated. (0 = Completely impossible, 10 = Extremely possible)	6 (5, 8)	7 (5, 9)	0.10
Confidence in COVID-19 vaccination’s ability to protect self from the COVID-19 virus after vaccination. (0 = Not confident at all, 10 = Extremely confident)	7 (6, 8)	8 (7, 9)	0.003
Confidence in COVID-19 vaccination’s ability to confer protection against the COVID-19 virus to loved ones and friends after their vaccination. (0 = Not confident at all, 10 = Extremely confident)	7 (6, 8)	8 (7, 9)	0.004
Self-perceived knowledge of the COVID-19 and its symptoms. (0 = Do not know much at all, 10 = Know extremely well)	8 (7, 8)	8 (7, 9)	0.054
Felt being coerced into COVID-19 vaccination, n (%)	21 (18)	8 (6)	0.006
**Trust in healthcare system and authorities, median (25th, 75th percentiles)**			
Perceived capability of Singapore’s medical system in handling another outbreak. (0 = Not capable at all, 10 = Extremely capable)	8 (7, 9)	8 (7, 9)	0.54
Trust in Singapore’s health authorities (e.g., Ministry of Health) in making decisions in the population’s best interest in terms of the COVID-19 vaccines provided. (0 = Do not trust at all, 10 = Extremely trust)	9 (7, 10)	9 (8, 10)	0.09
Perceived transparency of the authorities on information about safety of the COVID-19 vaccines. (0 = Not transparent at all, 10 = Extremely transparent)	7 (6, 9)	8 (7, 9)	0.002
**Risk/benefit**			
Does work require direct contact with COVID-19 patients?, n (%)			<0.001
*No*	109 (93)	93 (75)	
*Yes*	8 (7)	31 (25)	
Perceived eventual duration of the pandemic, n (%)			0.10
*1 year or less*	20 (17)	35 (28)	
*More than 1 year to 3 years*	77 (66)	67 (54)	
*3 or more years*	20 (17)	22 (18)	
Perceived severity of COVID-19 situation in Singapore, median (25th, 75th percentiles) (0 = Not severe at all, 10 = Extremely severe)	5 (3, 6)	5 (3, 7)	0.46
Perceived likelihood of loved ones and friends getting infected by COVID-19, median (25th, 75th percentiles) (0 = Not likely at all, 10 = Extremely likely)	6 (4, 8)	5.5 (4, 7)	0.78
Perceived time taken to recover from COVID-19 and its possible complications			0.274
*Unsure (if no symptoms experienced)*	12 (10)	12 (10)	
*2 weeks or less*	17 (15)	20 (16)	
*More than 2 weeks to 2 months*	26 (22)	40 (32)	
*More than 2 months to permanent*	62 (53)	52 (42)	
Perceived extent of life being affected once infected by COVID-19 virus, median (25th, 75th percentiles) (0 = Do not affect my life at all, 10 = Severely affects my life)	8 (7, 10)	8 (7, 10)	0.63
Degree of being emotionally affected (feeling afraid or scared) from the possibility of contracting the COVID-19 virus, median (25th, 75th percentiles) (0 = Not affected emotionally at all, 10 = Extremely affected emotionally)	8 (5, 9)	7 (5, 8)	0.61
**Immunization as a social norm**			
Voluntarily taken other vaccination (e.g., flu vaccine or Human Papilloma Virus) previously, n (%)	83 (71)	94 (76)	0.39
**Vaccine/Vaccination Specific Issues**			
Perceived likelihood of taking any vaccine (e.g., flu or Human Papilloma Virus) if offered free and at a convenient place, median (25th, 75th percentiles) (0 = Not likely at all, 10 = Extremely likely)	9 (8, 10)	10 (9, 10)	< 0.001
**Concerns on Vaccines characteristics, *n (%)***			
“I am concerned about the vaccine’s efficacy”			0.541
*Yes*	78 (66.7)	78 (62.9)	
*No*	39 (33.3)	46 (37.1)	
“Country in which vaccine is manufactured forms part of my concerns”			0.955
*Yes*	27 (23.1)	29 (23.4)	
*No*	90 (76.9)	95 (76.6)	
“I will decide against taking a particular COVID-19 vaccine if there is distrust in the manufacturer”			0.59
*Yes*	54 (46)	53 (43)	
*No*	63 (54)	71 (57)	
“I believe generally in vaccines”			0.111 ^#^
*No*	5 (4.3)	1 (0.8)	
*Yes*	112 (95.7)	123 (99.2)	
“l am worried about experiencing allergic reactions and/or anaphylaxis”			0.107
*Yes*	65 (55.6)	56 (45.2)	
*No*	52 (44.4)	68 (54.8)	
“I am worried about all side effects”			0.866
*Yes*	61 (52.1)	66 (53.2)	
*No*	56 (47.9)	58 (46.8)	
Long term side (LT) effects *			0.016
*Yes*	46 (43.4)	34 (28.1)	
*No*	60 (56.6)	87 (71.9)	
Sensitivity analyses LT side effects if missing data are all “no I do not have LT side effects concerns”			0.05
*Yes*	46 (39.3)	34 (27.4)	
*No*	71 (60.7)	90 (72.6)	
Sensitivity analyses LT side effects if missing data are all “yes I have LT side effects concerns”			0.003
*Yes*	57 (49)	37 (30)	
*No*	60 (51)	87 (70)	

* missing data n = 14, ^#^ Fisher’s exact test applied. SCH, SingHealth Community Hospitals.

**Table 3 vaccines-10-00537-t003:** Odds ratios of covariates on COVID-19 vaccination acceptance.

Covariates	Unadjusted OR (95% CI)	*p*-Value	Adjusted OR (95% CI)	*p*-Value
Male	2.67 (1.37–5.22)	0.004	2.44 (1.03–5.78)	0.043
Age (years)	1.03 (1.01–1.06)	0.014	1.06 (1.02–1.10)	0.002
Profession				
*Administrative*	Reference Group	-	-	-
*Allied health*	1.68 (0.85–3.33)	0.135	1.98 (0.82–4.75)	0.13
*Physician*	4.17 (1.75–9.9)	0.001	2.50 (0.82–7.58)	0.11
*Nurses*	4.32 (2.03–9.18)	<0.001	1.56 (0.52–4.64)	0.43
Nature of work being COVID-19 patient fronting	4.54 (1.99–10.4)	<0.001	2.38 (0.77–7.35)	0.13
Nationality				
*Non-Singaporeans*	Reference Group	-	-	-
*Singaporeans*	0.53 (0.30–0.91)	0.023	1.53 (0.56–4.10)	0.41
Living arrangement				
*Staying alone*	Reference Group	-	-	-
*Staying with family*	0.75 (0.26–2.15)	0.59	0.25 (0.06–1.07)	0.06
*Staying with friends, colleagues and/or housemates*	2.33 (0.69–7.84)	0.171	1.08 (0.23–4.99)	0.92
Presence of loved ones or friends infected with COVID-19	3.72 (1.74–7.98)	0.001	4.90 (1.68–14.27)	0.004
Felt coerced into getting COVID-19 vaccination	0.32 (0.13–0.74)	0.008	1.04 (0.32–3.44)	0.95
Transparency in information	1.26 (1.08–1.47)	0.003	1.19 (0.95–1.49)	0.13
Sources of COVID-19 vaccine information				
*Family and friends*	0.47 (0.23–0.95)	0.025	0.48 (0.18–1.26)	0.14
*Print Newspaper*	0.53 (0.30–0.92)	0.025	0.34 (0.16–0.72)	0.005
Belief in achieving herd immunity towards the COVID-19 virus even if he/she does not get vaccinated	0.88 (0.81–0.96)	0.003	0.92 (0.83–1.03)	0.14
Confidence in COVID-19 vaccination’s ability to protect self after vaccination	1.29 (1.1–1.51)	0.002	1.32 (0.86–2.03)	0.21
Confidence in COVID-19 vaccination’s ability to protect loved ones and friends after their vaccination	1.25 (1.07–1.45)	0.004	0.87 (0.59–1.28)	0.48
Perceived likelihood of taking any vaccine if offered free and at a convenient place	1.34 (1.14–1.57)	<0.001	1.12 (0.94–1.35)	0.21
Concerned about long term side effects from COVID-19 vaccination	0.51 (0.29–0.89)	0.017	0.78 (0.38–1.60)	0.50

OR, odds ratio. CI, confidence intervals.

## Data Availability

Data availability request may be sought from the corresponding author if needed.

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
