# Peer review of "COVID-19-Related Vaccine Hesitancy among Community Hospitals’ Healthcare Workers in Singapore"

_vaccines, 2022, doi:10.3390/vaccines10040537_

Round 1

Reviewer 1 Report

The article “COVID-19 related vaccine hesitancy among community hospitals’ healthcare workers in Singapore” is a well written study, with sound methodology and interesting results. The study methods and results are described in detail.

Minor suggestion

  • it appears that HCPs less than 40 yo were more hesitant.  Please analyze the groups again by using this age group as a reference group  (instead of only presenting age as a )and better determine the age group that needs to be targeted. The wording "younger age" is not clear and can be more specific in this way. If a statistically significant difference is found among age < 40 and Age >40 then please define the “younger age” in the abstract as well.

Author Response

Author's reply:

Please kindly see the Word file attachment.

Reviewer 2 Report

This is an interesting study on an important public health problem.

Please provide the following changes to improve the scientific soundness of this manuscript:

  1. The Introduction section is too short. Please provide a paragraph on COVID-19 in Singapore (epidemiology, major anti-epidemic measures that were in place; public acceptance of the lock-downs) - this would be important for international readers 
  2. The study aim should be clearly defined ("The aim of this study was....")
  3. Please provide the correct number of the sub-section (3. Results)
  4. The study group is relatively low. Please address this comment in discussion and the potential impact of the low study group on the generalization of the results.
  5. Please provide practical implications as well as further research needs.
  6. Conclusions should be more precise and balanced based on the limited number of the study group.

Author Response

(The authors gave the same response as above.)

Reviewer 3 Report

In the paper results of a survey concerning COVID-19 vaccine hesitancy are presented. Despite the fact that there is a number of papers analyzing reasons for which people decide to be vaccinated against COVID-19 or not, this paper is interesting and well written. The survey was conducted in a healtcare workers of three hospitals in Singapore. The Authors clearly described the studied group and the applied methodology. Also the results are described in detail and properly analyzed. Moreover, limitations of the study were correctly identified and some possibilities for further investigations were proposed. The obtained results of the described survey can help in planning possible future vaccination campaignes.

In my opinion the paper could be considered for publication in its present form.

Author Response

Thank you so much for your comments. We felt greatly encouraged by your inputs and will strive to make all of our future works relevant and useful to the scientific community.

Round 2

Reviewer 2 Report

The manuscript was improved in line with the suggestions. Thank you for the precise responses to the reviewers' comments.